# A population-based study of administrative data linkage to measure melanoma surgical and pathology quality

**Douglas R. McKay[1], Paul Nguyen[2], Ami Wang[3], Timothy P. Hanna [2,4,5] ***

**1** Division of Plastic Surgery, Department of Surgery, Queen's University, Kingston, Ontario, Canada, **2** ICES at Queen's University, Kingston, Ontario, Canada, **3** Department of Pathology and Molecular Medicine, Queen's University, Kingston, Ontario, Canada, **4** Department of Oncology, Queen's University, Kingston, Ontario, Canada, **5** Division of Cancer Care and Epidemiology, Cancer Research Institute at Queen's University, Kingston, Ontario, Canada

* tim.hanna@kingstonhsc.ca

## Abstract

**Data Availability Statement:** The dataset from this study is held securely in coded form at ICES. While data sharing agreements prohibit ICES from making the dataset publicly available, access may be granted to those who meet pre-specified criteria

### Background

Continuous quality improvement is important for cancer systems. However, collecting and compiling quality indicator data can be time-consuming and resource-intensive. Here we explore the utility and feasibility of linked routinely collected health data to capture key elements of quality of care for melanoma in a single-payer, universal health care setting.

### Method

This pilot study utilized a retrospective population-based cohort from a previously developed linked administrative data set, with a 65% random sample of all invasive cutaneous melanoma cases diagnosed 2007–2012 in the province of Ontario. Data from the Ontario Cancer Registry was utilized, supplemented with linked pathology report data from Cancer Care Ontario, and other linked administrative data describing health care utilization. Quality indicators identified through provincial guidelines and international consensus were evaluated for potential collection with administrative data and measured where possible.

### Results

A total of 7,654 cases of melanoma were evaluated. Ten of 25 (40%) candidate quality indicators were feasible to be collected with the available administrative data. Many indicators (8/25) could not be measured due to unavailable clinical information (e.g. width of clinical margins). Insufficient pathology information (6/25) or health structure information (1/25) were less common reasons. Reporting of recommended variables in pathology reports varied from 65.2% (satellitosis) to 99.6% (body location). For stage IB-II or T1b-T4a melanoma patients where SLNB should be discussed, approximately two-thirds met with a surgeon experienced in SLNB. Of patients undergoing full lymph node dissection, 76.2% had adequate evaluation of the basin.

for confidential access, available at www.ices.on.ca/DAS.

**Funding:** This study was supported by Kingston Health Sciences Centre Foundation (Fund 70174). Awarded to DM, TH, AW (www.uhkf.ca). The sponsor played no role in the study design, data collection and analysis, decision to publish, or preparation of the manuscript. This study was also supported by ICES, which is funded by an annual grant from the Ontario Ministry of Health and Long-Term Care (MOHLTC). The opinions, results and conclusions reported in this paper are those of the authors and are independent from the funding sources. No endorsement by ICES or the Ontario MOHLTC is intended or should be inferred. T.P.H. holds a research chair provided by the Ontario Institute for Cancer Research through funding provided by the Government of Ontario (#IA-035).

**Competing interests:** The authors have declared that no competing interests exist.

## Conclusions

We found that use of linked administrative data sources is feasible for measurement of melanoma quality in some cases. In those cases, findings suggest opportunities for quality improvement. Consultation with surgeons offering SLNB was limited, and pathology report completeness was sub-optimal, but was prior to routine synoptic reporting. However, to measure more quality indicators, text-based data sources will require alternative approaches to manual collection such as natural language processing or standardized collection. We recommend development of robust data platforms to support continuous re-evaluation of melanoma quality indicators, with the goal of optimizing quality of care for melanoma patients on an ongoing basis.

## Introduction

Concerns regarding surgical treatment quality for melanoma have been raised in Canada and other jurisdictions [1–3]. A population-based study in the Canadian province of Ontario suggests significant variations exist in melanoma surgical management, potentially leading to inadequate care for some, with greater likelihood of inadequate treatment with advanced age, female sex, and in certain jurisdictions [4]. At the same time, stage and outcome have been reported to vary by patient factors such as age, sex and area-level socioeconomic status, suggesting a need to optimize the quality of care between groups to ensure optimal outcomes [2, 3].

Surgery and pathology quality improvement is essential. High quality surgery and pathology reporting are necessary to guide stage-based treatment decisions for optimal melanoma care, including choice of adjuvant systemic therapy that can improve cancer outcomes, including survival [5–8]. Improving the quality of care can also reduce waste, and improve the cost-effectiveness of care for melanoma [9]. Notably, formally developed quality indictors validated through expert consensus exist to assess melanoma care quality, though they have not been explored as broadly as for other cancers [10].

Continuously measuring and reporting on quality indicators for melanoma and other cancers is crucial, as it may contribute to reducing inappropriate practice variation, and improving outcomes [11]. However, collecting and compiling the information required to do so can be resource intensive, time consuming and costly. We thus set out to explore the utility and feasibility of linked routinely collected health data to capture key elements of the quality of surgical management and pathologic reporting in a single payer, universal health setting. To do so, we set out to describe melanoma quality of care in Ontario, using a well-characterized population-based sample with detailed pathology report information [3]. Linked administrative data could provide a means to efficiently and continuously measure variation in quality of care, to improve the quality of care for those diagnosed with melanoma.

## Methods

This pilot study utilized a previously developed retrospective population-based cohort. A 65% random sample of all invasive melanoma cases diagnosed in Ontario between January 1, 2007 and December 31, 2012 that were recorded in the Ontario Cancer Registry (OCR) was used for this study. The random sample was a convenience sample based on power requirements for the parent study investigating melanoma treatment outcomes according to stage in Ontario

(CIHR MOP 137022) [3]. This source was chosen for the pilot study because completely abstracted population-based pathology data for later years was not available, and we wished to assess feasibility using the most comprehensive population-level data currently available. The OCR is administered by Ontario Health Cancer Care Ontario (CCO), the provincial cancer agency associated with Ontario's single payer universal health system. The random sample thus provides insights into the management of the complete population of people diagnosed with melanoma over the study period. This study was approved by the Queen's University Health Sciences and Affiliated Teaching Hospitals Research Ethics Board (EPID-425-13). This study followed the Strengthening the Reporting of Observational Studies in Epidemiology (STROBE) reporting guideline for cohort studies.

In the case of multiple primaries, details from the earliest melanoma were used. Patients whose first diagnosis was pure *in situ* disease were excluded based on concerns that OCR *in situ* data was incomplete. Patients without a pathology report from CCO and with core/FNA only biopsies were excluded. Out-of-province residents treated in Ontario and Ontario residents without continuous provincial medical coverage in a 5-year look back from diagnosis were excluded as were those under 20 years of age. OCR data was utilized to identify cases of melanoma. OCR demonstrates a high level of population coverage, including for melanoma [12, 13]. Available pathology reports for all patients were provided from CCO and abstracted according to a standardized algorithm developed at the Queen's University Division of Cancer Care and Epidemiology, and deterministically linked to each patient's OCR record according to their group ID, a unique identifier. Reliability testing indicated 97% complete agreement between all three abstractors and a clinician experienced in melanoma for primary variables, including stage-defining items. M-category data was supplemented by stage information provided by regional cancer centers.

Rurality was measured via the Rurality Index for Ontario (RIO). The RIO score is based on a 0–100 scale, with higher scores indicating a greater degree of rurality [14]. Comorbidity was measured using the Elixhauser comorbidity index with 5-year lookback from diagnosis, based on Canadian Institute for Health Information Discharge Abstract Database and Same Day Surgery data [15].

To pilot the use of linked administrative data to measure quality indicators, we sought out a comprehensive source that provided a representative selection of quality metrics deemed important by many clinicians for measuring melanoma surgery and pathology quality. We also determined a-priori that these metrics should be contemporaneous to the administrative data utilized for this pilot project. Sources were sought out based on literature searches and expert knowledge of the study team. One key study was identified, that identified quality indicators via a comprehensive, iterative consensus-based process involving thirteen experts, based on review of the literature and consensus guidelines [10]. Face, construct and predictive validity were considered in their process. These quality indicators were supplemented by non-overlapping quality metrics identified by review of consensus-based contemporaneous provincial guidelines for the province of Ontario [16].

Quality indicators fell into three broad categories: (1) Pathology reporting of disease, (2) Management of the sentinel node, and (3) Management of the nodal basin. These quality indicators were reviewed and evaluated for their feasibility of assessment with the administrative data linked to the available pathology report data. Features of the administrative data required for measuring the quality indicator, and a description of the type of data affecting measurement feasibility was recorded. Where relevant, the type of missing data affecting the feasibility of measurement with administrative data was catalogued (clinical, pathologic or structural, where 'structural' refers to details of the organization or structure of the health care system).

## Results

### Study cohort

We identified a cohort of 7,654 adult patients with invasive melanoma, provincial health coverage and linked pathology records from 8,877 linkable patients with 9,462 pathology records (S1 Appendix, Table 1). The most common reasons for exclusion were missing pathology records at first diagnosis (n = 438), in situ melanoma as first diagnosis (n = 386) and lapses in provincial health coverage (n = 289). The latest primary site pathology reports for each patient were reviewed. Of these, 5,146 (67.2%) described an excision, 1,649 (21.5%) described a biopsy, 77 (1.0%) described an amputation. The procedure type was missing or not reported for 782 (10.2%) of cases.

### Quality indicator measurement feasibility with linked administrative data

There were 25 non-overlapping consensus-based quality indicators identified (Table 2). Ten of 25 (40%) candidate quality indicators were feasible to be collected with the administrative data. Many indicators (8/25) could not be measured due to unavailable clinical information in administrative data sources (e.g. width of clinical margins) (Table 2). Lack of sufficient pathology information (6/25) or health structure information (1/25) were less common reasons for unfeasibility.

### Quality indicator measurement

The quality indicators that were feasible for measurement with administrative data linkage, and the relevant patient numbers meeting them are provided in Table 3. Additional details are provided in the subsequent paragraphs.

### Pathology reporting

**QI 1: Completeness of primary site pathology report.** 3,802 (49.7%) patients reported variables required by the quality indicator (Table 3), with 3,933 (51.4%) complete if Clark level was excluded, and 4,271 (55.8%) if Clark level and mitosis were excluded. High levels of reporting were noted for Breslow thickness and Clark level of invasion at 99.1% and 94.9%, respectively. Other variables are summarized in Table 4 and range down to 2.6% for in transit metastasis documentation.

### Management of the sentinel node quality indicators

Amongst the 7,654 patients, 1,012 (13.2%) patients had sentinel lymph node biopsy (SLNB) procedures, within 6 months of diagnosis.

**QI 2: Appropriate certification of surgeons performing SLNB or LND.** There were 1,189 patients undergoing SLNB or LND in the study cohort. All operating surgeons were confirmed to be certified by the College of Physicians and Surgeons of Ontario.

**QI 3: Referral for SLNB discussion Stage IB-II.** Stage IB-II melanoma patients were included in this analysis as were T1b-T4a patients. All patients with documented distant metastases (i.e., M1) were excluded as were those with dermal deposits as the appropriate role of SLNB is not well defined. Amongst the first primary of the 7,654 patients, 4,132 (54.0%) satisfied these conditions. Of these, 2,726 (66.0%) consulted with a surgeon who had previously performed SLNB or lymphoscintigraphy plus another lymph node procedure on the basis of their previous OHIP billings (S2 Appendix) within 6 months of their diagnostic date. Compared to those who consulted with a SLNB surgeon, the 1,406 (34.0%) who had not consulted were older (median [IQR]: 67 [54–79] vs. 62 [51–74], $p<0.001$), more often resided in rural

**Table 1. Patient and disease characteristics of first primary melanoma (N = 7,654).**

| Patient and Disease Characteristics | Total |
|---|---|
| Year of diagnostic date | |
| 2007 | 1,179 (15.40%) |
| 2008 | 1,212 (15.83%) |
| 2009 | 1,329 (17.36%) |
| 2010 | 1,336 (17.45%) |
| 2011 | 1,402 (18.32%) |
| 2012 | 1,196 (15.63%) |
| Age at diagnostic date | |
| Mean ± SD | 61.90 ± 16.02 |
| Median (IQR) | 63 (51–75) |
| Age (categorized) at diagnostic date | |
| 20–39 | 729 (9.52%) |
| 40–49 | 1,024 (13.38%) |
| 50–59 | 1,526 (19.94%) |
| 60–69 | 1,710 (22.34%) |
| 70–79 | 1,506 (19.68%) |
| 80+ | 1,159 (15.14%) |
| Sex | |
| Female | 3,575 (46.71%) |
| Male | 4,079 (53.29%) |
| Neighbourhood income quintile at diagnostic date | |
| Missing | 19 (0.25%) |
| 1 (Lowest) | 1,051 (13.73%) |
| 2 | 1,322 (17.27%) |
| 3 | 1,467 (19.17%) |
| 4 | 1,712 (22.37%) |
| 5 (Highest) | 2,083 (27.21%) |
| Rurality Index for Ontario (RIO) at diagnostic date | |
| Missing/NA | 75 (0.98%) |
| Urban (0≤RIO<10) | 4,621 (60.37%) |
| Suburban (10≤RIO<40) | 2,158 (28.19%) |
| Rural (40≤RIO) | 800 (10.45%) |
| Elixhauser comorbidity index (5-year lookback from diagnostic date) | |
| 0 | 5,969 (77.99%) |
| 1 | 837 (10.94%) |
| 2–3 | 578 (7.55%) |
| 4+ | 270 (3.53%) |
| Histology | |
| Melanoma, NOS | 2,448 (31.98%) |
| Nodular | 990 (12.93%) |
| Superficial spreading | 3,099 (40.49%) |
| Acral lentiginous | 124 (1.62%) |
| Desmoplastic | 66 (0.86%) |
| Lentigo maligna | 555 (7.25%) |
| Spindle cell melanoma, NOS | 60 (0.78%) |
| Malignant melanoma in giant pigmented nevus | 74 (0.97%) |
| Other | 238 (3.11%) |

(*Continued*)

**Table 1.** (Continued)

| Patient and Disease Characteristics | Total |
|---|---:|
| Body location | |
| Missing | 31 (0.41%) |
| Head and neck | 1,482 (19.36%) |
| Upper trunk | 1,000 (13.07%) |
| Lower trunk | 511 (6.68%) |
| Trunk or back, NOS | 1,007 (13.16%) |
| Arm or shoulder | 1,919 (25.07%) |
| Leg or hip | 1,688 (22.05%) |
| Other | 16 (0.21%) |
| Laterality | |
| Missing | 1,253 (16.37%) |
| Left | 3,236 (42.28%) |
| Right | 2,953 (38.58%) |
| Midline | 212 (2.77%) |
| Minimum AJCC 7th edition best stage* | |
| IA | 3,045 (39.78%) |
| IB | 2,239 (29.25%) |
| IIA | 657 (8.58%) |
| IIB | 525 (6.86%) |
| IIC | 361 (4.72%) |
| IIIA | 200 (2.61%) |
| IIIB | 263 (3.44%) |
| IIIC | 279 (3.65%) |
| IV | 85 (1.11%) |

Notes:

* Minimum best stage based on derived pT, pN and pM stage classifications.

areas (based on RIO scores) (13.8% vs. 8.8%, $p<0.001$) and had a history of higher comorbidities ($\geq2$ comorbidities: 13.8% vs. 10.5%, $p = 0.002$).

Of the 2,726 who saw these surgeons, 638 (23.4%) had a SLNB with 270 having positive nodes. Compared those who had SLNB, the 2,088 (76.6%) patients who saw these surgeons and did not have SLNB were older (median [IQR]: 63 [51–75] vs. 61 [50–72], $p<0.001$) with no significant differences for residency in rural areas (8.3% vs. 10.3%, $p = 0.273$) and history of greater comorbidities ($\geq2$ comorbidities: 11.1% vs. 8.6%, $p = 0.078$).

**QI 4: Referral for SLNB discussion for high risk $<1.0$ mm.** 1,548 (20.2%) patients with melanomas $<1.0$ mm in thickness had high risk features: 498 (32.2%) were young ($\leq39$ years), 1,157 (74.7%) with $\geq1$ mm$^2$ mitotic rate, 77 (5.0%) with ulceration, and 23 (1.5%) with a positive deep margin from a shave biopsy. 647 (41.8%) consulted with a SLNB surgeon within 6 months of diagnosis. 61 of 647 had a SLNB procedure with 25 having positive nodes found.

**QI 5: Use of lymphoscintigraphy.** Of the 1,012 patients of any stage who had a SLNB, 473 (46.7%) had at least one positive node and 103 (10.2%) had more than one nodal drainage basin dissected. Lymphoscintigraphy was used in 909 (89.8%). Lymphoscintigraphy procedures have increased compared to the earliest years in the cohort (82.4% 2007, up to 90.3% in 2012).

**Table 2. Established surgical and pathology melanoma quality indicators and feasibility of collection with administrative data.**

| Source | Quality Indicator | Quality Domain | Feasible with Available Administrative Data | Comments | Missing data type |
|---|---|---|---|---|---|
| Bilimoria et al [10], Quality Indicators selected via consensus-based process | If a surgeon performs SLNB or LND for melanoma, then the surgeon must be certified by the American Board of Surgery or equivalent board or international association. | Structure | Yes | Based on linked provincial physician database indicating presence of College of Physicians and Surgeons of Ontario practice number | NA |
| | If a patient has a melanoma in situ (ie, Tis), then the surgical excision margins must be 5 mm (or the specific anatomic or cosmetic factors that limit margin distance should be noted). | Process | No | Clinical excision margins not collected as operative reports not collected. These are now increasingly available through province-wide digital networks. Also, incomplete cancer registry data on melanoma in situ | Clinical |
| | If a patient has a melanoma, then the surgeon must document the measured surgical margin in the operative report. | Process | No | Clinical excision margins not collected as operative reports not collected. These are now increasingly available through province-wide digital networks. | Clinical |
| | If a patient has a melanoma, then a clear histologic margin must be documented. | Process | No | The provincial cancer registry only collects pathology reports where cancer is reported. Wide excision reports with clear margins and no invasive cancer were thus not available to confirm clear margins. | Pathology |
| | If a patient has a melanoma ≤ 1 mm thick (ie, T1), then the surgical excision margins must be 1 cm (or the specific anatomic or cosmetic factors that limit margin distance should be noted). | Process | No | Clinical excision margins not collected as operative reports not collected. These are now increasingly available through province-wide digital networks. | Clinical |
| | If a patient has a melanoma 1–2 mm thick (ie, T2), then the surgical excision margins must be 1–2 cm (or the specific anatomic or cosmetic factors that limit margin distance should be noted). | Process | No | Clinical excision margins not collected as operative reports not collected. These are now increasingly available through province-wide digital networks. | Clinical |
| | If a patient has a melanoma ≥ 2 mm thick (ie, T3 or T4), then the surgical excision margins must be 2–3 cm (or the specific anatomic or cosmetic factors that limit margin distance should be noted). | Process | No | Clinical excision margins not collected as operative reports not collected. These are now increasingly available through province-wide digital networks. | Clinical |
| | If a patient is to undergo an SLNB, then lymphoscintigraphy must be performed to identify the draining nodal basin(s). | Process | Yes | Via hospital-reported procedure data and linked physician reimbursement data. | NA |
| | If a patient undergoes an SLNB, then the SLNs must be sent for permanent sectioning only (no frozen sections), unless a reason is documented. | Process | No | Pathology service reimbursement data incomplete for the study period. | Pathology |
| | If a patient undergoes an SLNB, then the SLNs must be examined with serial sectioning/HE and with IHC if the HE analysis is negative or equivocal (ie, S-100, HMB-45, and MART-1). | Process | No | Data not collected during pathology data abstraction. Possible to collect if source variable is specified. | Pathology |
| | If a patient has clinically apparent/palpable lymphadenopathy, then an LND must not be performed without an antecedent histologic diagnosis. | Process | No | Data on pre-LND biopsy results not collected during pathology data abstraction. Possible to collect if source variable is specified. | Pathology |
| | If a patient has a stages Ib or II melanoma, SLNB must be discussed with the patient. | Process | Yes* | The indicator can be measured, with the provision that consultation with a surgeon who performs SLNB can be determined, but administrative records do not capture the content of the consultation discussion. | NA |
| | If a patient undergoes a cervical LND or CLND, then at least 15 regional lymph nodes must be resected and pathologically examined. | Process | Yes | Using linked pathology report data. | NA |
| | If a patient undergoes an axillary LND or CLND, then at least 10 regional lymph nodes must be resected and pathologically examined. | Process | Yes | Using linked pathology report data. | NA |
| | If a patient undergoes an inguinal LND or CLND, then at least five regional lymph nodes must be resected and pathologically examined. | Process | Yes | Using linked pathology report data. | NA |
| | If a patient has a melanoma, then the pathology report must document Breslow thickness, Clark level, histologic ulceration, peripheral/radial and deep margin statuses, satellitosis, anatomic location of the lesion, regression, and mitotic rate. | Process | Yes | Using linked pathology report data. | NA |
| | If a patient undergoes an SLNB or LND for melanoma, then the pathology report must document the number of lymph nodes examined and the number of lymph nodes found to contain metastases. | Process | Yes | Using linked pathology report data. | NA |
| | If a patient has clinically palpable nodal disease of the inguinofemoral nodes, then a pelvic CT or PET must be obtained to rule out pelvic lymphadenopathy. | Process | Yes | Using pathology report data linked to physician reimbursement data. | NA |
| | If a patient with melanoma has biopsy-proven or palpable nodal disease and no evidence of distant metastases, then the patient must undergo a LND. | Process | No | Administrative sources could not identify all patients meeting this criteria for undergoing LND. | Clinical |
| | If a patient has a metastatic lymph node detected on SLNB, then a CLND must be performed except in the context of a clinical trial or if the patient has severe comorbidities. | Process | No | Incomplete information on clinical trials in administrative data. | Clinical |
| | If a patient with melanoma has biopsy-proven, palpable nodal disease, then the patient should not undergo SLNB. | Process | No | Data on pre-LND biopsy results not collected. Possible to collect if source variable is specified. | Pathology |

*(Continued)*

**Table 2.** (Continued)

| Source | Quality Indicator | Quality Domain | Feasible with Available Administrative Data | Comments | Missing data type |
|---|---|---|---|---|---|
| Ontario Health Cancer Care Ontario. Wright et al [16]. | SLNB should be discussed with patients with melanomas <1.0 mm in thickness and with high-risk features, such as young age, mitotic rate ≥1 mm2, ulceration, or diagnosis by shave biopsy if the deep margin is positive and consequently the depth of the lesion may be underestimated | Process | Yes* | The indicator can be measured, with the provision that consultation with a surgeon who performs SLNB can be determined, but administrative records do not capture the content of the consultation discussion. | NA |
| | Standard synoptic pathology reporting should be used | Process | No | Content of synoptic reports not captured in pathology database. | Pathology |
| | Intradermal injection of radioactive tracer and either patent blue or lymphazurin blue dye is recommended. | Process | No | Information on injection of dye not available in administrative sources. | Clinical |
| | SLNB should be performed only following discussion of the options with the patient, in a unit with access to appropriate surgical, nuclear medicine and pathology services. | Process | No | Administrative data missing elements required to evaluate appropriateness of surgical, nuclear medicine and pathology services | Structural |

*indicates a provision to the measurement of the quality indicator is specified in the comments.

**Table 3. Achievement of quality indicators (QIs) first primary melanoma (N = 7,654).**

| Quality Indicators | Total |
|---|---|
| **Pathology Reporting** | |
| QI 1: IF a patient has a melanoma, THEN the pathology report must well-document all Breslow thickness, Clark level, histologic ulceration, peripheral/radial and deep margin status, satellitosis, anatomic location of the lesion, regression, and mitotic rate. | 3,802 (49.67%) |
| **Management of the Sentinel Node** | |
| QI 2: If a surgeon performs SLNB or LND for melanoma, then the surgeon must be certified by the American Board of Surgery or equivalent board or international association. | 1,189 (100.00%)* |
| QI 3: IF a patient has a Stage IB or II melanoma, SLNB must be discussed with the patient. | 2,726 (65.97%) |
| QI 4: SLNB should be discussed with patients with melanomas <1.0 mm in thickness and with high-risk features, such as young age, mitotic rate ≥1 $mm^2$, ulceration, or diagnosis by shave biopsy if the deep margin is positive and consequently the depth of the lesion may be underestimated | 647 (41.80%) |
| QI 5: IF a patient is to undergo a SLNB, THEN lymphoscintigraphy must be performed to identify the draining nodal basin(s) when drainage to more than one basin is possible. | 909 (89.82%) |
| QI 6a: IF a patient first undergoes a SLNB for melanoma, THEN the pathology report must document the number of lymph nodes examined and the number of lymph nodes found to contain metastases. | >1,006 (100.00%)** |
| **Management of the Nodal Basin** | |
| QI 6b: IF a patient first undergoes a LND for melanoma, THEN the pathology report must document the number of lymph nodes examined and the number of lymph nodes found to contain metastases. | 190 (100.00%) |
| QI 7: IF a patient undergoes a cervical LND or CLND, THEN at least 15 regional lymph nodes must be resected and pathologically examined. | 46 (66.67%) |
| QI 8: IF a patient undergoes an axillary LND or CLND, THEN at least 10 regional lymph nodes must be resected and pathologically examined. | 101 (77.69%) |
| QI 9: IF a patient undergoes an inguinal LND or CLND, THEN at least 5 regional lymph nodes must be resected and pathologically examined. | 61 (82.43%) |
| QI 10: IF a patient has clinically palpable nodal disease of the inguinofemoral nodes, THEN a pelvic CT or PET must be obtained to rule out pelvic lymphadenopathy | 38 (71.70%) |

Notes:

* Number of patients undergoing SLNB or LND.

** Exact numbers cannot be provided due to privacy regulations for groups of patients of five or less.

**Table 4. Reporting of pathologic variables of first primary melanoma (N = 7,654).**

| Pathologic Characteristics | Total |
|---|---|
| Non-missing and applicable responses | |
| Invasion | 7,654 (100.00%) |
| Body location | 7,623 (99.59%) |
| Laterality | 6,401 (83.63%) |
| Breslow thickness (including minimal thickness) | 7,587 (99.12%) |
| Clark Level | 7,262 (94.88%) |
| Mitotic rate | 5,908 (77.19%) |
| Ulceration | 7,137 (93.25%) |
| Lymphovascular invasion | 5,506 (71.94%) |
| Tumor infiltrating lymphocytes | 4,960 (64.8%) |
| Perineural invasion and/or neurotropism | 3,886 (50.77%) |
| Presence of regression | 5,408 (70.66%) |
| Dermal deposit | 290 (3.79%) |
| Satellites or microsatellites | 4,987 (65.16%) |
| In transit metastases | 196 (2.56%) |

QI 6a: Lymph node reporting for SLNB.   Almost all of the 1,012 patients had underwent SLNB as their first procedure (>1006). Of these patients, 473 (47%) had at least 1 positive node found, with the majority having one (343) or two (101) positive nodes. One-hundred percent reported on the number of lymph nodes examined and the number of lymph nodes found to contain metastases.

### Management of the lymph node basin quality indicators

QI 6b: Lymph node reporting for non-sentinel lymph node (NSLN) dissection.   Of the 190 patients whose first procedure within 6 months of diagnosis was NSLN, 144 (78.7%) had at least 1 positive node found. The majority (124) of these patients had 1–6 positive nodes extracted from NSLN. One-hundred percent (190) reported on the number of lymph nodes examined and the number of lymph nodes found to contain metastases.

QI 7: Cervical LND or CLND node count.   69 patients underwent NSLN or regional nodal recurrence resection of the neck. Of these, 45 (65.2%) had only a NSLN clearance of the neck and 24 (34.8%) had a regional nodal recurrence resected (plus NSLN clearance in <6 cases). 46 (66.7%) of 69 patients undergoing neck node surgery had at least 15 regional lymph nodes removed and assessed. Of those 24 patients presenting with regional neck nodal recurrence, >18 had at least 15 lymph nodes assessed.

QI 8: Axillary LND or CLND node count.   130 patients had procedures in the axilla. This group includes NSLN and the resection of regional nodal recurrences. Of the 130 patients, 101 (77.7%) had at least 10 regional lymph nodes removed and assessed. 84 only underwent NSLN and 46 underwent a resection for regional recurrence (plus NSLN clearance in <6 cases). For the 84 NSLN patients, 71 (84.5%) had at least 10 regional lymph nodes assessed; in the case of regional recurrence, 30 (65.2%) had at least 10 nodes assessed.

QI 9: Inguinal LND or CLND node count.   74 patients had either only NSLN (46) or recurrence dissections (28, with <6 also having NSLN clearance). Of the 74 patients, 61 (82.4%) had at least 5 regional lymph nodes assessed. For the 46 NSLN patients, >40 had at least 5 nodes assessed.

QI 10: Radiologic staging for clinical inguinal/femoral adenopathy.   53 patients had a NSLN procedure in the absence of a prior SLNB or regional nodal resection at the inguinal site. Abdominal and pelvic CT scans were identified using OHIP billing codes (S2 Appendix); 38 (71.7%) patients had a pelvic CT scan within the 6 months prior to the nodal surgery date.

## Discussion

In this population-based sample, we evaluated the feasibility of using linked administrative data sources to measure the quality of melanoma surgery and pathology reporting on the basis of consensus-derived quality indicators and provincial evidence-based guidelines. We found that a minority of quality indicators could be measured with the available linked data sources (10/25). Missing clinical information was the most common reason quality indicators could not be measured (8/25). In our pilot cohort, we identified variation in surgery and pathology reporting according to quality indicators relevant to practice at that time (2007–2012). These findings provide a strong impetus to develop ways to continuously measure and improve a comprehensive suite of quality indicators for melanoma in our region, and beyond.

We note that since the time of our pilot cohort, some metrics may have improved due to adoption of synoptic reporting in the province [17, 18]. Ulceration, when present, draws into question the depth of the primary and could not be determined in 517 (6.8%) of patients. Only 4,987 (65.2%) patients reported any information on the presence of satellites or microsatellites. Re-evaluation of these indicators will provide insights into the effectiveness of synoptic

reporting in ensuring the quality and completeness of pathology reporting in the setting of a universal single-payer health system.

For other metrics, it is not as certain that improvements have been made since the time of our pilot cohort. For example, SLNB and the beneficial role of lymphoscintigraphy for melanoma have been well documented in the literature for almost 20 years [19–22]. The final analysis of the first Multicenter Selective Lymphadenectomy Trial (MSLT-1) only became available in 2014 confirming the benefits of SLNB; however, reports on the benefits of SLNB in this trial span the era under study [23–25]. By considering whether the surgeon seen in consultation had experience performing SLNB based on previous billing code use, we saw that only 2,726 (66.0%) of Stage IB-II and T1b-T4a patients eligible for a sentinel node biopsy were seen by such a surgeon, and less for high-risk <1mm patients (647). Nevertheless, we hypothesize that SLNB referral has increased due to availability of new effective adjuvant targeted and immune-based therapies where pathologically confirmed lymph node involvement is required for funding. The additional impact of MSLT-II and the German Dermatologic Cooperative Oncology Group (DeCOG) SLNB trial on SLNB rates are less clear. These studies suggest that completion dissection increases regional disease control and provide prognostic information for SLNB positive patients, without increasing melanoma-specific survival (MSLT-II) or distant metastasis-free survival (DeCOG) [26, 27]. Whether this evidence supporting SLNB alone without completion dissection for SLNB positive patients would increase use of SLNB by possibly removing hesitance to use SLNB is unclear. This requires further confirmatory study.

Stage-based treatment cannot be executed without an adequate summary of the extent of disease. Of all 4,132 potential SLNB patients (Stage IB-II and T1b-T4a), only 638 underwent a SLNB, a procedure known to be a powerful prognosticator. Patient choice may play a role, but the possibility of inadequate access is concerning. We observed that patients seen by a SLNB surgeon, and those subsequently receiving SLNB tended to be younger, and have lower comorbidity in univariate analysis. Whether these differences reflect patient-centered decision making cannot be determined based on the available data. We also observed that eligible patients living in rural areas were less likely to see a SLNB surgeon. Again it is unclear whether this reflects patient choice, poor geographic access or both.

Furthermore, the number of lymph nodes dissected for non-SLNB procedures failed to meet established quality metrics for between 18% (inguinal basin) to 33% (cervical basin) of cases. The question of adequacy of resection is not novel to this study [21–24]. Spillane et al in a retrospective review of the number of nodes taken at the Melanoma Institute Australia (MIA) demonstrated that over 90% of cases had greater or equal to 7 and 10 nodes in the specimen for the groin and axilla respectively. For the neck they subdivided results on the basis of levels taken: for those cases where 3 or less levels were excised greater than 90% of patients had 6 or more nodes taken and for neck dissections of four or more levels greater than 90% of patients has 20 or more nodes taken [22]. There were similar findings in Rossi et al's large patient series from nine Italian Melanoma Intergroup institutions [28]. If we consider the MIA as a center of expertise and directly compare provincial data on the number of nodes resected, outcomes fall below the proposed 90% threshold number of nodes taken for inguinal and axilla basins. Unlike the MIA data, our cervical basin data is not stratified by levels taken, though only 67% had at least 15 nodes resected. We note that others have found similar underperformance for LND/CLND benchmarks in large population data. For example Bilimoria et al found achievement of the LND/CLND consensus-based benchmarks were low in the United States National Cancer Database [10]. Adequacy of surgical management of these basins bears further scrutiny for these populations.

There is no prospective data for melanoma linking these numbers to recurrence-free survival, but the numbers themselves can be used as a proxy for thoroughness of treatment and

pathologic evaluation. Regular evaluation of lymph node dissection quality remains important in the MSLT-II era. This study showed no melanoma-specific-survival advantage for patients who went on to regional node resection following positive sentinel node biopsy compared to nodal observation with ultrasonography [26]. Importantly, SLNB positive patients now managed with ultrasound follow-up rather than CLND will commonly develop lymph node recurrence (26% by five years in MSLT-II), and for these patients, the quality of salvage lymph node dissection bears importance. Measurement of appropriate imaging follow-up for SLNB positive patients without CLND will also be an important quality indicator going forward.

To support a program of quality improvement for melanoma pathology and surgery, continuous measurement and reporting of a comprehensive suite of melanoma quality indictors is required. Though the available quality metrics showed important gaps in quality of care requiring attention, we importantly found that only a minority of quality metrics could be measured with meticulously collected linked administrative data sources. How can this be addressed? With the rise of electronic synoptic reporting, some information sources are being more efficiently and completely collected in our jurisdiction and many others. However, other important variables require attention. For example, clinical margin width is only collected systematically in the individual patient chart. Manual abstraction for all patients is time- and resource-intensive. Other alternatives are routine sampling of a small, representative subset of patients from each institution on an ongoing basis. A sampling approach has been used in radiation oncology quality measurement, though it still required a substantial amount of time and coordination [29]. Artificial intelligence (AI) with natural language processing may provide a solution for routinely collected pathology and surgery variables [30]. This requires a comprehensive data infrastructure and human capacity to ensure the training of algorithms to ensure complete and accurate data capture.

There are limitations to our study. Some metrics that are relevant to the clinical community may not have been included in the recommended consensus-based quality indicators we evaluated. For example, the false negative rate after SLNB or SLNB positivity rate. We note that the former requires multiple data sources and extended follow-up, though is technically feasible with appropriate data linkage. The latter is also feasible, without extended follow-up. We also note that recommendations on the optimal number of lymph nodes to be dissected varies to some degree between studies, though the purpose of our present investigation was to explore feasibility and utility of using data linkage for quality measurement, rather than to determine the optimal metric [28, 31].

The pilot cohort under study provided a unique opportunity to study melanoma quality of care at the population level, as we had access to detailed pathologic data linked to administrative sources on a random sample of all cases in the Canadian province of Ontario. However, the sample was historic, covering 2007 to 2012. This was the pre-MSLT-II era, and the use of completion dissection has now changed. However, as noted, it is possible that more patients will require salvage dissection due to nodal recurrence due to use of nodal observation rather than completion dissection, emphasizing the need to ensure the quality of nodal evaluation and follow-up for melanoma. Though historic, our pilot sample provides far more comprehensive pathology data than available in a typical cancer registry. Nonetheless, only 10/25 (40%) of quality indicators were feasibly measured with the linked administrative data. Six of the indicators not measurable with administrative data related to margins as we did not have access to surgical reports documenting the clinical excision margin (5/6 metrics), or pathology report information for excision samples with no residual melanoma that would have confirmed negative margins in many cases (1/6 metrics). Our findings emphasize the need to improve processes for cancer patient data reporting, abstraction and linkage, to ensure that a comprehensive suite of quality indicators can be collected on a continual basis for patients with melanoma, and other cancers.

## Conclusion

In this-population-based pilot study of a universal health care system, we found that there was limited feasibility of using linked administrative data sources for measuring quality indicators. For the 40% of indicators that were feasibly measured, evidence suggests opportunities for quality improvement in surgical and pathological quality of care. Completeness of pathologic reporting was sub-optimal in this historic cohort and this may be improved by synoptic reporting. Consultations with surgeons offering SLNB occurred in only two thirds of eligible cases, and far less received SLNB, limiting staging and prognostic information relevant for adjuvant therapy. Nearly one quarter of patients had less than the optimal number of nodes in their pathology report in non-SLNB dissection. We recommend development of robust data platforms to support continuous re-evaluation of melanoma quality indicators in a contemporary sample of patients, with the goal of optimizing quality of care for melanoma patients on an ongoing basis.

## Supporting information

**S1 Appendix. Identification of invasive cutaneous melanoma patients and their pathology records in Ontario from January 1, 2007 to December 31, 2012.**
(DOCX)

**S2 Appendix. Ontario Health Insurance Plan (OHIP) billing codes.**
(DOCX)

## Acknowledgments

Parts of this material are based on data and/or information compiled and provided by CIHI. However, the analyses, conclusions, opinions and statements expressed in the material are those of the author(s), and not necessarily those of CIHI. Parts of this material are based on data and information provided by Ontario Health Cancer Care Ontario (CCO). The opinions, results, view, and conclusions reported in this paper are those of the authors and do not necessarily reflect those of CCO. No endorsement by CCO is intended or should be inferred. These data were linked using unique encoded identifiers and analyzed at ICES.

## Author Contributions

**Conceptualization:** Douglas R. McKay, Ami Wang, Timothy P. Hanna.

**Data curation:** Paul Nguyen.

**Formal analysis:** Paul Nguyen.

**Funding acquisition:** Douglas R. McKay, Ami Wang, Timothy P. Hanna.

**Investigation:** Paul Nguyen, Timothy P. Hanna.

**Methodology:** Douglas R. McKay, Paul Nguyen, Ami Wang, Timothy P. Hanna.

**Project administration:** Timothy P. Hanna.

**Resources:** Douglas R. McKay, Timothy P. Hanna.

**Software:** Paul Nguyen.

**Supervision:** Paul Nguyen, Timothy P. Hanna.

**Validation:** Douglas R. McKay, Paul Nguyen, Ami Wang, Timothy P. Hanna.

**Writing – original draft:** Ami Wang, Timothy P. Hanna.

**Writing – review & editing:** Douglas R. McKay, Paul Nguyen, Ami Wang, Timothy P. Hanna.

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
