## [Decision Letter · Decision Letter 0]

15 Nov 2021

PONE-D-21-29441A Population-Based Study of Administrative Data Linkage to Measure Melanoma Surgical and Pathology QualityPLOS ONE

Dear Dr. Hanna,

Thank you for submitting your manuscript to PLOS ONE. After careful consideration, we feel that it has merit but does not fully meet PLOS ONE’s publication criteria as it currently stands. Therefore, we invite you to submit a revised version of the manuscript that addresses the points raised during the review process.

We look forward to receiving your revised manuscript.

Kind regards,

Sandro Pasquali, M.D., Ph.D.

Academic Editor

PLOS ONE

Journal Requirements:

"This study was supported by ICES, which is funded by an annual grant from the Ontario Ministry of Health and Long-Term Care (MOHLTC). The opinions, results and conclusions reported in this paper are those of the authors and are independent from the funding sources. No endorsement by ICES or the Ontario MOHLTC is intended or should be inferred. Parts of this material are based on data and/or information compiled and provided by CIHI. However, the analyses, conclusions, opinions and statements expressed in the material are those of the author(s), and not necessarily those of CIHI. Parts of this material are based on data and information provided by Cancer Care Ontario (CCO). The opinions, results, view, and conclusions reported in this paper are those of the authors and do not necessarily reflect those of CCO. No endorsement by CCO is intended or should be inferred. These data were linked using unique encoded identifiers and analyzed at ICES. 

T.P.H. holds a research chair provided by the Ontario Institute for Cancer Research through funding provided by the Government of Ontario (#IA-035)"

"This study was supported by Kingston Health Sciences Centre Foundation (Fund 70174). Awarded to DM, TH, AW. www.uhkf.ca. The sponsor played no role in the study design, data collection and analysis, decision to publish, or preparation of the manuscript"

Additional Editor Comments (if provided):

The Authors did a great job in improving their manuscript after previous submission. Here are somme comments that are meant to highlight findings in the discussion section.

Given their expertise, can Authors provide in the discussion information about implications for clinical practice and research of their findings?

Among missing information, histopathologic margins are very important. Can you please comment on this missing information?

How these results compared with existing literature?

How Authors expect their results to be in a more contemporary patient series? For instance, do they expect rate of SLNB performance even lower after results of the German trial and MSLT-II? Which are the implications for adjuvant therapies? it would be interesting to have Authors view.

The main comment is about the final conclusion of the manuscript. There are differences between abstract and full-text which should be report consistent information. I would rather reccoment to use the conclusion of the full-text manuscript which look more informative to me.

Reviewers' comments:

Reviewer's Responses to Questions

**Comments to the Author**

1. Is the manuscript technically sound, and do the data support the conclusions?

Reviewer #1: Yes

2. Has the statistical analysis been performed appropriately and rigorously? 

Reviewer #1: Yes

3. Have the authors made all data underlying the findings in their manuscript fully available?

Reviewer #1: Yes

4. Is the manuscript presented in an intelligible fashion and written in standard English?

Reviewer #1: Yes

5. Review Comments to the Author

Reviewer #1: The authors have performed substantial changes in the text, especially in the introduction and discussione section, as required by my previous review. The limits of the study are clearly stated and the implications for further improvements in the topic of quality of care in melanoma provided.

No further changes are needed

6. PLOS authors have the option to publish the peer review history of their article (what does this mean?). If published, this will include your full peer review and any attached files.

Reviewer #1: No

---

## [Author Response · Author response to Decision Letter 0]

5 Jan 2022

Our response to Reviewers is provided as an attached Word document labeled "Cover Letter"

---

## [Editor Report · Decision Letter 1]

26 Jan 2022

A population-based study of administrative data linkage to measure melanoma surgical and pathology quality

PONE-D-21-29441R1

Dear Dr. Hanna,

We’re pleased to inform you that your manuscript has been judged scientifically suitable for publication and will be formally accepted for publication once it meets all outstanding technical requirements.

Kind regards,

Sandro Pasquali, M.D., Ph.D.

Academic Editor

PLOS ONE

Additional Editor Comments (optional):

Sydney Melanoma Unit (SMU) - this is now called Melanoma Institute Australia, just change it please.
---

## [Editor Report · Acceptance letter]

9 Feb 2022

PONE-D-21-29441R1 

A population-based study of administrative data linkage to measure melanoma surgical and pathology quality 

Dear Dr. Hanna:

I'm pleased to inform you that your manuscript has been deemed suitable for publication in PLOS ONE. Congratulations! Your manuscript is now with our production department. 

Kind regards, 

on behalf of

Dr. Sandro Pasquali 

Academic Editor

PLOS ONE